# FLOW GRAPH NEURAL NETWORKS

## ABSTRACT

Graph Neural Networks (GNNs) have become essential for learning from graph-structured data. However, existing GNNs do not consider the conservation law inherent in graphs associated with a flow of physical resources, such as electrical current in power grids or traffic in transportation networks. To address this limitation and enhance the performance on tasks where accurate modeling of resource flows is crucial, we propose Flow Graph Neural Networks (FlowGNNs). This novel GNN framework adapts existing graph attention mechanisms to reflect the conservation of resources by distributing a node's message among its outgoing edges instead of allowing arbitrary duplication of the node's information. We further extend this framework to directed acyclic graphs (DAGs), enabling discrimination between non-isomorphic flow graphs that would otherwise be indistinguishable for standard GNNs tailored to DAGs. We validate our approach through extensive experiments on two different flow graph domains—electronic circuits and power grids—and demonstrate that the proposed framework enhances the performance of traditional GNN architectures on both graph-level classification and regression tasks.

## 1 INTRODUCTION

Graph-structured data represents the complex relationships and interactions between entities as a set of nodes and edges and is prevalent across many real-world domains, such as social networks (Fan et al., 2019), recommender systems (Wu et al., 2022), materials science (Reiser et al., 2022) or epidemiology (Liu et al., 2024). Traditional deep learning methods, which are typically designed for Euclidean data such as images (Li et al., 2021) or sequences (Lim & Zohren, 2021), fail to fully exploit the irregular structure of graphs. To address this, graph neural networks (GNNs) (Scarselli et al., 2008; Kipf & Welling, 2017) have emerged as a powerful framework that extends the scope of deep learning to graph-based data, enabling models to learn both node-level features as well as the underlying graph topology through iterative message-passing between neighboring nodes. As graph data becomes increasingly common, advancing GNN architectures is crucial for improving performance in tasks such as node classification (Hamilton et al., 2017), graph regression (Gilmer et al., 2017), or link prediction (Zhang & Chen, 2018).

In many important applications of GNNs, graphs are naturally associated with a flow of physical resources, such as electrical current in electronic circuits (Sánchez et al., 2023) or power grids (Liao et al., 2021), traffic in transportation networks (Jiang & Luo, 2022), water in river networks (Sun et al., 2021), or raw materials and goods in supply chains (Kosasih & Brintrup, 2022). In these *(resource) flow graphs*, all nodes, except for source and target nodes, are subject to Kirchhoff's first law, which states that the sum of all incoming and outgoing flows must be zero, reflecting the conservation of resources. By contrast, *informational graphs*—such as computation graphs, social networks, or citation networks—are not associated with any physical flow but rather represent relationships or information transfer. Information can be arbitrarily duplicated and propagated in these graphs, unlike in flow graphs, where such duplication would violate the conservation law.

As a result, two non-isomorphic graphs may be *equivalent* as informational graphs (e.g., they represent the same computation) but *non-equivalent* as flow graphs (e.g., they represent different electronic circuits). An example of this is given in Fig. 1. Since the result of the sine operation can be duplicated without constraints and transmitted to arbitrarily many other operations, the two non-isomorphic graph structures represent the same computation. However, the two graph structures may also represent electronic circuits, which are governed by Kirchhoff's first law. In this case, the two

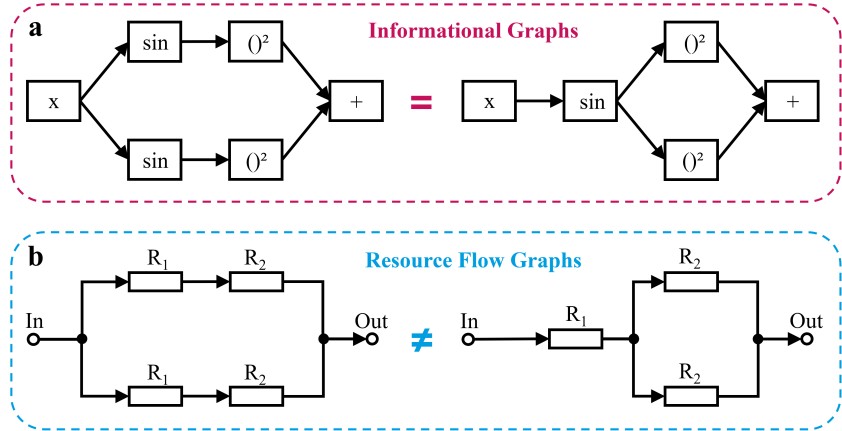

Figure 1: Two non-isomorphic graphs that are *equivalent* in the case of informational graphs, but *different* as resource flow graphs. **a** The two different directed graph structures represent the same computation (example adapted from Zhang et al. (2019)). **b** The same graph structures as above represent different electronic circuits.

circuits are *different*. Although they can be transformed into each other's graph structure by combining or splitting resistors, this would lead to different resistances, i.e., node features. In this case, a sufficiently expressive GNN should be able to map the graph structures to different representations.

In recent years, many new GNN models have been specifically designed for different graph types (Thomas et al., 2023). However, despite their fundamental differences, informational graphs and flow graphs are still treated by the same basic message-passing layers (MPLs), such as GCN (Kipf & Welling, 2017), GIN (Xu et al., 2019) or GAT (Veličković et al., 2018). In these models, messages exchanged between neighboring nodes do not depend on the number of message recipients. Instead, the information is *arbitrarily duplicated* and passed to all neighbors. Even attention mechanisms, as applied in GAT, GATv2 (Brody et al., 2022) or Graph Transformer (Shi et al., 2021), are only normalizing across *incoming* messages and therefore cannot overcome this limitation.

Many flow graphs, including the example graphs in Fig. 1, can be naturally expressed as directed acyclic graphs (DAGs), e.g., operational amplifiers (Dong et al., 2023) or material flow networks (Perera et al., 2018). In these cases, nodes are typically updated sequentially following the partial order of the DAG, and the final target node representation is used as the graph embedding (Zhang et al., 2019; Thost & Chen, 2021). However, since directed acyclic GNNs are utilizing non-conservational message-passing schemes resulting in identical target node representations, they are not capable of distinguishing between non-isomorphic flow graphs such as in Fig. 1.

A possible approach to overcome the problem of indistinguishable flow graphs is to use node indices or random features as input node features (Loukas, 2020; Sato et al., 2021), which makes the model capable of uniquely identifying each node. However, the resulting GNN model is no longer permutation invariant, which reduces its generalization capability. Similar problems arise for Transformer-based models (Vaswani et al., 2017) such as PACE (Dong et al., 2022), which incorporate the relational inductive bias Battaglia et al. (2018) via positional encodings. A different strategy would be to introduce Kirchhoff's first law through an additional physics-informed loss term (Donon et al., 2020), which is useful if the target variable is the resource flow itself. However, introducing additional loss terms considerably increases the training complexity and does not overcome the fundamental limitations of message-passing neural networks in distinguishing non-equivalent flow graphs.

To overcome the above problems that arise when applying message-passing (directed acyclic) GNNs to resource flow graphs, we propose a new GNN framework that builds upon attentional GNNs. Instead of normalizing the attention scores across incoming neighbors, we normalize them across *outgoing* neighbors. This simple but effective modification ensures that the message of a specific

node is distributed among all message recipients and thereby avoids arbitrary message duplication, reflecting the conservation of physical resources in flow graphs.

Our contributions are the following:

1. **GNN framework for flow graphs:** We develop a new framework called Flow Graph Neural Network (FlowGNN) which replaces the standard attention mechanism of existing GNNs with a flow attention mechanism that ensures the conservation of physical resources as they traverse through the graph.

2. **GNN model for directed acyclic flow graphs:** We further extend the new framework to DAGs, resulting in a model called Directed Acyclic Flow Graph Neural Network (DAFlowGNN): We show that DAFlowGNN can distinguish non-isomorphic directed acyclic flow graphs which would otherwise be mapped to the same representation by standard DAGNNs.

3. **Extensive Experiments:** We conduct experiments on two different flow graph domains (electronic circuits and power grids), covering both undirected graphs and DAGs, and show that our proposed models outperform their standard counterparts on graph-level classification and regression tasks across multiple datasets.

The code is available at `https://anonymous.4open.science/r/FlowGNN-24`.

## 2 Preliminaries

**Graph** A directed graph can be defined as a tuple $\mathcal{G} = (\mathcal{V}, \mathcal{E})$ containing a set of nodes $\mathcal{V} \subset \mathbb{N}$ and a set of directed edges $\mathcal{E} \subseteq \mathcal{V} \times \mathcal{V}$. Thereby, we define $e = (u, v)$ as the *directed edge* from node $v$ to node $u$. An edge is called *undirected* if $(u, v) \in \mathcal{E}$ whenever $(v, u) \in \mathcal{E} \; \forall \, u, v \in \mathcal{V}$. Furthermore, we call the set $\mathcal{N}_{\text{in}}(v) = \{u \in \mathcal{V} \mid (v, u) \in \mathcal{E}\}$ the *incoming neighborhood* of $v$ and the set $\mathcal{N}_{\text{out}}(v) = \{u \in \mathcal{V} \mid (u, v) \in \mathcal{E}\}$ the *outgoing neighborhood* of $v$.

**Directed Acyclic Graph** A graph $\mathcal{G}$ is *cyclic*, if there exists a subgraph $\mathcal{H} = (\{v_1, \ldots, v_k\}, \{e_1, \ldots, e_k\}) \subseteq \mathcal{G}, \, v_i \in \mathcal{V}, \, e_i \in \mathcal{E} \; \forall \, i$, such that the sequence of nodes and edges $v_1, e_1, v_2, e_2, \ldots, v_k, e_k, v_1$ is a closed path of length $k$ with $v_i \neq v_j \; \forall \, v_i, v_j$. Otherwise, it is called *acyclic*. A *directed acyclic graph* (DAG) is a graph that is directed and acyclic. In the context of DAGs, we also call the incoming neighborhood the *predecessors* of a node, and the outgoing neighborhood the *successors* of a node. The set of all *ancestors* of node $v$ contains all nodes $u \in \mathcal{V}$ such that $v$ is reachable from $u$. Similarly, the *descendants* are the nodes $u \in \mathcal{V}$ that are reachable from $v$. Finally, the set of nodes without predecessors is called the set of *start* or *initial nodes*, denoted by $\mathcal{I} \subset \mathcal{V}$, and the set of nodes without successors is called the set of *end* or *final nodes*, denoted by $\mathcal{F} \subset \mathcal{V}$.

**Flow Graph** Let $\mathcal{S}, \mathcal{T} \subseteq \mathcal{V}$ be two fixed subsets of $\mathcal{V}$ (the sources and targets of $\mathcal{V}$). A *flow* on $\mathcal{G}$ is a mapping $f : \mathcal{E} \to \mathbb{R}$ that satisfies Kirchhoff's first law:

$$\sum_{u \in \mathcal{N}_{\text{in}}(v)} f(v, u) = \sum_{u \in \mathcal{N}_{\text{out}}(v)} f(u, v) \quad \forall \, u \in \mathcal{V} \setminus \{\mathcal{S}, \mathcal{T}\}. \tag{1}$$

If a graph is associated with a flow $f$ as defined above, we refer to it as a *flow graph*. In DAGs, the start nodes are sources and the end nodes are targets: $\mathcal{I} \subseteq \mathcal{S}$ and $\mathcal{F} \subseteq \mathcal{T}$.

### 2.1 Graph Neural Networks

Graph Neural Networks (GNNs) transfer the concept of traditional neural networks to graph data. Thereby, the node representations $\{\boldsymbol{h}_i \in \mathbb{R}^F \mid i \in \mathcal{V}\}$ with the feature dimension $F$ are updated iteratively by aggregating information from neighboring nodes via message-passing. The updated node representations $\{\boldsymbol{h}_i' \in \mathbb{R}^F \mid i \in \mathcal{V}\}$, i.e., the output of the network layer, are given by

$$\boldsymbol{h}_i' = \phi \left( \boldsymbol{h}_i, \bigoplus_{j \in \mathcal{N}_{\text{in}}(i)} \psi(\boldsymbol{h}_j) \right), \tag{2}$$

with a learnable message function $\psi$, an aggregation scheme $\oplus$, e.g., sum or mean, as well as an update function $\phi$. The choice of $\phi$, $\oplus$, and $\psi$ are defining the design of a specific GNN model.

## 2.2 ATTENTIONAL GRAPH NEURAL NETWORKS

An *attentional* GNN layer takes a set of input node features $\{\boldsymbol{h}_i \in \mathbb{R}^F \mid i \in \mathcal{V}\}$ and uses a scoring function $e : \mathbb{R}^F \times \mathbb{R}^F \to \mathbb{R}$ to compute attention coefficients

$$e_{ij} = e\left(\boldsymbol{h}_i, \boldsymbol{h}_j\right) \tag{3}$$

that indicate the importance of the features of node $j$ to node $i$. Popular attentional GNNs include GAT (Veličković et al., 2018), GATv2 (Brody et al., 2022) and Graph Transformer (GT) (Shi et al., 2021), which mainly differ in the choice of the scoring function $e$. We briefly discuss these models in App. A.1.

The computed attention coefficients $e_{ij}$ are normalized across all incoming neighboring nodes $j$ using the softmax function:

$$\alpha_{ij} = \text{softmax}_j(e_{ij}) = \frac{\exp(e_{ij})}{\sum_{k \in \mathcal{N}_{\text{in}}(i)} \exp(e_{ik})}. \tag{4}$$

Note that, in general, $\alpha_{ij} \neq \alpha_{ji}$ for undirected edges due to the normalization, even if the same attention scores $e_{ij} = e_{ji}$ are assigned to these two edges, such as in GAT or GATv2. The hidden states of node $i$ are finally updated using a non-linearity $\sigma$:

$$\boldsymbol{h}_i' = \sigma\left(\sum_{j \in \mathcal{N}_{\text{in}}(i)} \alpha_{ij} \boldsymbol{W} \boldsymbol{h}_j\right). \tag{5}$$

The standard graph attention mechanism is visualized in Fig. 2a.

## 2.3 DIRECTED ACYCLIC GRAPH NEURAL NETWORKS

The main idea of directed acyclic GNNs is that the nodes are processed and updated sequentially according to the partial order defined by the DAG. Thereby, the update of a node representation $\boldsymbol{h}_i$ is computed based on the current-layer node representations of node $i$'s predecessors. Consequently, the message-passing for a node can only be carried out if all of its predecessors' hidden representations have already been computed, which is only possible because the underlying graph is acyclic. The message-passing scheme of directed acyclic GNNs can therefore be expressed as

$$\boldsymbol{h}_i' = \phi\left(\boldsymbol{h}_i, \bigoplus_{j \in \mathcal{N}_{\text{in}}(i)} \psi\left(\boldsymbol{h}_j'\right)\right). \tag{6}$$

The most widely used directed acyclic GNNs are D-VAE (Zhang et al., 2019) and DAGNN (Thost & Chen, 2021). These models utilize gated recurrent units (GRU) as the update function $\phi$ and are briefly explained in App. A.2. As an alternative to sequential models, DAGs can also be encoded using Transformer-based architectures, such as PACE (Dong et al., 2022).

## 3 FLOWGNN MODELS

### 3.1 FLOW GRAPH ATTENTIONAL LAYER

The problem with standard attention mechanisms, when applying them to flow graphs, is that the attention scores are normalized across all *incoming* edges. Therefore, a message from node $j$ to node $i$ does not depend on how many nodes this message is passed to, and thus, non-equivalent flow graphs as in Fig. 1 are not distinguishable. To fix this problem, we propose to normalize the attention scores across *outgoing* edges instead (see Fig. 2b). We call the resulting weights *flow attention weights* and denote them as $\beta_{ij}$ in order to distinguish them from the standard attention weights $\alpha_{ij}$:

$$\beta_{ij} = \text{softmax}_i(e_{ij}) = \frac{\exp(e_{ij})}{\sum_{k \in \mathcal{N}_{\text{out}}(j)} \exp(e_{kj})}. \tag{7}$$

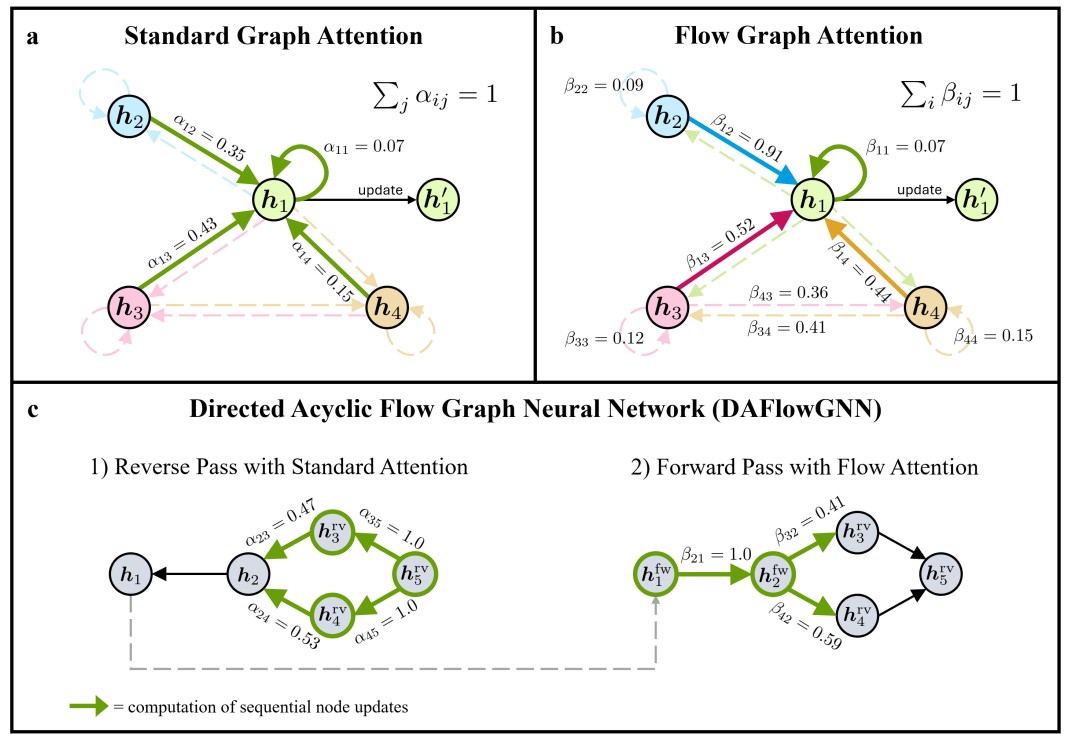

Figure 2: **a** Standard graph attention mechanism as it is applied in attentional GNNs. The attention weights associated with edges of the same color sum to 1. **b** The proposed flow attention mechanism applied in FlowGNNs. The flow attention weights associated with edges of the same color sum to 1. **c** Two snapshots during the reverse and forward pass of the Directed Acyclic Flow Graph Neural Network (DAFlowGNN). Nodes marked in green have already been updated.

Although the attention scores are normalized across outgoing edges, we still aggregate incoming messages in order to update the hidden state of node $i$:

$$\boldsymbol{h}_i' = \sigma\left(\sum_{j \in \mathcal{N}_{\text{in}}(i)} \beta_{ij} \boldsymbol{W} \boldsymbol{h}_j\right). \tag{8}$$

However, since the messages are multiplied with the flow attention weights $\beta_{ij}$, they now also depend on the neighborhood of the message's sender, i.e., node $j$. In this way, we ensure that a message transmitted by any node cannot be duplicated arbitrarily but instead is distributed among all outgoing neighbors. We define a *flow graph attentional layer* as the message-passing layer described in Eq. 8 and *flow graph neural networks (FlowGNNs)* as the family of attentional GNNs, which use one or more flow graph attentional layers with an arbitrary scoring function. Furthermore, we denote the corresponding FlowGNN versions of standard attentional GNNs as FlowGAT, FlowGATv2, FlowGT, etc.

### 3.2 DIRECTED ACYCLIC FLOWGNN

Directed acyclic GNNs map two non-isomorphic DAGs to the same representation as long as they represent the same computation (Zhang et al., 2019). However, we are interested in flow graphs rather than computational graphs. Therefore, we need to ensure that *all* non-isomorphic DAG structures are mapped to different representations. For this purpose, we propose a directed acyclic FlowGNN (DAFlowGNN), which builds upon DAGNN and incorporates the flow attention mechanism.

A naive approach to a FlowGNN for DAGs would be to start from a DAGNN and then replace the attention weights $\alpha_{ij}$ with flow attention weights $\beta_{ij}$. When computing these flow attention weights

associated with the outgoing edges from node $j$, we only have information about all ancestors of $j$, because we are updating the nodes according to the partial order of the DAG. However, since the flow of some arbitrary physical resource from node $j$ to node $i$ in a flow graph may also depend on all descendants of the node $i$, we should also include information about these descendants in the computation of the $\beta_{ij}$.

Therefore, we construct a DAFlowGNN layer from two sublayers (see Fig. 2c). In the first sublayer (we call it the *reverse pass*), we invert all edges of the DAG $\mathcal{G}$ and apply a standard DAGNN layer to the reverse DAG $\tilde{\mathcal{G}}$. This is equivalent to performing the aggregation over all successor nodes in the original DAG $\mathcal{G}$ instead of over all predecessor nodes:

$$\boldsymbol{m}_i^{\text{rv}} = \sum_{j \in \mathcal{N}_{out}(i)} \alpha_{ij}\left(\boldsymbol{h}_i, \boldsymbol{h}_j^{\text{rv}}\right) \boldsymbol{h}_j^{\text{rv}}, \tag{9}$$

$$\alpha_{ij}\left(\boldsymbol{h}_i, \boldsymbol{h}_j^{\text{rv}}\right) = \underset{j \in \mathcal{N}_{out}(i)}{\text{softmax}} \left((\boldsymbol{w}_1^{\text{rv}})^{\text{T}} \boldsymbol{h}_i + (\boldsymbol{w}_2^{\text{rv}})^{\text{T}} \boldsymbol{h}_j^{\text{rv}}\right), \tag{10}$$

$$\boldsymbol{h}_i^{\text{rv}} = \text{GRU}(\boldsymbol{h}_i, \boldsymbol{m}_i^{\text{rv}}). \tag{11}$$

In the second sublayer, we perform a *forward pass* on the original DAG $\mathcal{G}$. However, this time we are applying the flow attention mechanism described in Section 3.1 to compute flow attention weights $\beta_{ij}$:

$$\boldsymbol{m}_i^{\text{fw}} = \sum_{j \in \mathcal{N}_{out}(i)} \beta_{ij}\left(\boldsymbol{h}_i^{\text{rv}}, \boldsymbol{h}_j^{\text{fw}}\right) \boldsymbol{h}_j^{\text{fw}}, \tag{12}$$

$$\beta_{ij}\left(\boldsymbol{h}_i^{\text{rv}}, \boldsymbol{h}_j^{\text{fw}}\right) = \underset{i \in \mathcal{N}_{out}(j)}{\text{softmax}} \left((\boldsymbol{w}_1^{\text{fw}})^{\text{T}} \boldsymbol{h}_i^{\text{rv}} + (\boldsymbol{w}_2^{\text{fw}})^{\text{T}} \boldsymbol{h}_j^{\text{fw}}\right), \tag{13}$$

$$\boldsymbol{h}_i^{\text{fw}} = \text{GRU}(\boldsymbol{h}_i^{\text{rv}}, \boldsymbol{m}_i^{\text{fw}}). \tag{14}$$

Since the hidden states $\boldsymbol{h}_i^{\text{rv}}$ of the reverse pass contain information about all descendants of the node $i$, and the hidden states $\boldsymbol{h}_j^{\text{fw}}$ contain information about all ancestors of the node $j$, the computation of the flow attention weights $\beta_{ij}$ essentially takes into account information about all nodes of the graph that are connected to the node $i$.

After $L$ DAFlowGNN layers, we compute the graph-level representation from both the reverse pass representations of the start nodes as well as the forward pass representations of the end nodes and concatenate across layers:

$$\boldsymbol{h}_{\mathcal{G}} = \underset{i \in \mathcal{I}}{\text{Max-Pool}} \left( \overset{L}{\underset{l=0}{\big\|}} \boldsymbol{h}_i^{\text{rv},l} \right) \big\| \underset{j \in \mathcal{F}}{\text{Max-Pool}} \left( \overset{L}{\underset{l=0}{\big\|}} \boldsymbol{h}_j^{\text{fw},l} \right). \tag{15}$$

The separation of the DAFlowGNN layer into a reverse and a forward pass is necessary due to the sequential nature of the message-passing in GNNs for DAGs. Note that this architecture is not required in the undirected setting, because all nodes are updated simultaneously in this case. Therefore, the "forward" and "reverse" passes are performed at the same time and the computation of the flow attention weights always takes into account information about descendants *and* ancestors up to a distance defined by the number of FlowGNN layers. Finally, from a computational point of view, a DAFlowGNN layer is twice as expensive to compute compared to a DAGNN layer, due to the additional reverse pass. Therefore, the DAGNN model should have twice as many layers compared to the DAFlowGNN for a fair comparison of both models.

### 3.3 EXPRESSIVITY OF DAFLOWGNN

Consider the DAGs from Fig. 1. We can prove that any directed acyclic GNN, e.g., D-VAE or DAGNN, cannot distinguish between those two DAG structures by drawing the rooted subtrees of the end nodes. A rooted subtree visualizes the information flow through the graph, or in other words, the message-passing history that results in the node representation update of the end node. A standard message-passing GNN can only distinguish between two non-isomorphic node neighborhoods if the node's rooted subtrees are different (Xu et al., 2019). However, flow attention weights enable the distinction of non-isomorphic node neighborhoods despite identical rooted subtrees.

| DAG | Rooted Subtrees with example (flow) attention weights | | |
|---|---|---|---|
| | **D-VAE** | **DAGNN** | **DAFlowGNN** |
|  |  |  |  |
|  |  |  |  |

Figure 3: Rooted subtrees with example (flow) attention weights generated by different directed acyclic GNNs for two non-isomorphic directed acyclic flow graphs. While D-VAE is not calculating any attention weights, DAGNN is using a standard attention mechanism and DAFlowGNN is using the proposed flow attention mechanism. Flow attention weights that are different for the two DAGs are highlighted in bold. Node colors indicate different node features.

Fig. 3 shows the subtrees rooted at the green end nodes for both graph structures generated by different directed acyclic GNNs. Thereby, colors indicate different node features. We also add example (flow) attention weights to the corresponding edges, where applicable, which can be viewed as an additional option for distinguishing graphs. For all models, the two subtrees are structurally identical, so the only option to distinguish the two graphs would be the attention weights. Since D-VAE does not compute any attention weights, it maps the two graph structures to the same representation. The attention weights computed by DAGNN are *always* identical for both graph structures, since their sum over all incoming neighbors is equal to 1. Note that the attention weights corresponding to the incoming edges of the green end node depend on the features of the red and orange nodes, respectively. However, they are not affected by the different structures of the graphs. The only model capable of distinguishing the two DAGs is DAFlowGNN. Instead of normalizing the attention scores across incoming neighbors, it normalizes them across outgoing neighbors, resulting in different flow attention weights for the two DAGs.

## 4 EXPERIMENTS

### 4.1 DATASETS, TASKS, AND BASELINES

**Datasets** We perform experiments on two different flow graph datasets. First, we test different FlowGNNs (FlowGAT, FlowGATv2, and FlowGT) on publicly available power grid data from the PowerGraph dataset (Varbella et al., 2024), which encompasses the IEEE24, IEEE39, IEEE118, and UK transmission systems. The graphs contained in these datasets are undirected and cyclic and represent test power systems with the aim of mirroring real-world power grids. The test systems differ from each other in scale and topology, covering a wide range of relevant parameters. Secondly, we test DAFlowGNN on the Ckt-Bench101 dataset from the publicly available Open Circuit Benchmark (OCB) (Dong et al., 2023), which contains 10,000 operational amplifiers (Op-Amps) represented as DAGs. The dataset further provides circuit specifications for each Op-Amp, e.g., gain and bandwidth, which were obtained from circuit simulations. Further details on all datasets can be found in App. A.3.

**Tasks** For the PowerGraph dataset, we train the models to perform cascading failure analysis. Thereby, we utilize the attributed graphs provided by the PowerGraph dataset, each representing unique pre-outage operating conditions along with a set of outages corresponding to the removal of a single or multiple branches. An outage may result in demand not served (DNS) by the grid, and

a cascading failure may occur, meaning that one or more additional branches trip after the initial outage. In this scenario, we focus on two graph-level tasks: Binary and multiclass classification. For binary classification, the model is supposed to predict whether the grid is stable (DNS = 0 MW) or unstable (DNS > 0 MW) after the outage. For multiclass classification, the model should additionally predict whether a cascading failure occurs, resulting in four distinct categories representing the possible combinations of stable/unstable and cascading failure yes/no. For Ckt-Bench101, we perform graph-level regression to predict the properties of the Op-Amps. For this purpose, we train three separate instances of each model for the prediction of gain, bandwidth, and figure of merit (FoM), respectively. The FoM is a measure of the circuit's overall performance and depends on gain, bandwidth, and phase margin.

**Baselines** We compare the FlowGNNs on the PowerGraph dataset against their corresponding standard GNN versions GAT, GATv2, and GraphTransformer. Furthermore, we compare them against two more widely adopted non-attentional GNNs from the literature: GCN, and GINe (Hu et al., 2020), a modified version of GIN, which is able to incorporate edge features. In the second experiment, we compare DAFlowGNN against D-VAE, DAGNN, and PACE. As additional baselines, we further compare to standard GNNs and FlowGNNs not explicitly tailored to DAGs: GCN, GIN, GAT, GATv2, GT, FlowGAT, FlowGATv2, and FlowGT.

## 4.2 CASCADING FAILURE ANALYSIS ON POWER GRIDS

**Experimental setting** We train three different FlowGNNs (FlowGAT, FlowGATv2, FlowGT) and all baseline models for each test system contained in the PowerGraph dataset. Furthermore, we train all models on binary and multiclass classification as described in Sec. 4.1. We stick closely to the original benchmark setting in Varbella et al. (2024) by splitting the datasets into train/validation/test with ratios 85/5/10% and using the Adam optimizer (Kingma, 2014) with an initial learning rate of $10^{-3}$ as well as a scheduler that reduces the learning rate by a factor of five if the validation set accuracy stops improving for ten epochs. The negative log-likelihood is used as the loss function and balanced accuracy (Brodersen et al., 2010) is used as the evaluation accuracy due to the strong class imbalance (see App. A.3). We train all models with a batch size of 16 for a maximum number of 500 epochs but stop training with a patience of 20 epochs. Each model is trained with varying numbers of message-passing layers (1, 2, 3) with a hidden dimension of 32. Between subsequent message-passing layers, we apply the ReLU activation function followed by a dropout of 10%. In order to obtain graph embeddings from the node embeddings, we apply a global maximum pooling operator as the readout layer. As a final prediction layer, we use a single linear layer or a two-layer perceptron with a LeakyReLU activation function in between, depending on which type of prediction layer was used for the corresponding model in the original PowerGraph benchmark. Each individual training run is repeated five times with different random seeds.

**Discussion** The balanced accuracies on the test set are reported for each model on each of the four test systems for binary and multiclass classification in Tab. 1 and Tab. 2, respectively. First of all, we notice that the accuracy improves with more message-passing layers, which has already been observed for power grid data in Ringsquandl et al. (2021). Therefore, we only report the results for three layers here, while the results for one and two layers can be found in App. A.4. The FlowGNNs outperform their corresponding standard GNN version in the majority of cases: In both, binary and multiclass classification, FlowGAT shows a higher balanced accuracy compared to GAT for the test systems IEEE39 and IEEE118, and only a minimal performance decrease on the other test systems. In the case of GATv2, the FlowGNN version even outperforms its standard counterpart on all test systems, while for the transformers, FlowGT performs better than GT on all test systems except for IEEE118. These results indicate that the flow attention mechanism, which is the only applied change to the corresponding baselines, may be beneficial when working with flow graph data.

Across all tasks and test systems, GIN turns out to be the strongest baseline. Since GIN is a non-attentional GNN, our proposed flow attention mechanism cannot be incorporated. However, it still seems to perform well on flow graphs, which could be explained by the fact that it is a maximally expressive GNN (Xu et al., 2019). For binary classification, FlowGNNs outperform GIN on two of four test systems, while for multiclass classification, they outperform GIN on three of four test systems. Thereby, FlowGT achieves the highest accuracy among all models on the IEEE24 test system, while FlowGATv2 shows the highest accuracy on IEEE118 as well as on IEEE39 in the case of multiclass classification.

Table 1: Binary classification results for the cascading failure analysis on the PowerGraph dataset using three MPLs for all models. Reported results represent the balanced accuracy on the test set in %, averaged over five training runs with different random seeds, along with the corresponding standard deviation. The best result for each test system is marked in bold.

| Model | IEEE24 | IEEE39 | IEEE118 | UK |
|---|---|---|---|---|
| GCN | $91.3 \pm 2.2$ | $89.9 \pm 2.5$ | $86.0 \pm 4.8$ | $93.8 \pm 1.7$ |
| GIN | $98.1 \pm 0.9$ | $\mathbf{97.1 \pm 0.4}$ | $99.7 \pm 0.2$ | $\mathbf{98.8 \pm 0.9}$ |
| GAT | $94.7 \pm 1.4$ | $93.9 \pm 2.1$ | $92.1 \pm 10.5$ | $97.5 \pm 0.4$ |
| GATv2 | $91.8 \pm 1.8$ | $90.3 \pm 1.7$ | $90.5 \pm 9.8$ | $97.5 \pm 0.4$ |
| GT | $96.9 \pm 0.7$ | $95.6 \pm 1.4$ | $99.5 \pm 0.3$ | $97.7 \pm 0.2$ |
| FlowGAT | $94.0 \pm 1.4$ | $95.6 \pm 1.4$ | $99.4 \pm 0.3$ | $97.4 \pm 0.4$ |
| FlowGATv2 | $97.1 \pm 0.6$ | $96.8 \pm 1.0$ | $\mathbf{99.8 \pm 0.1}$ | $97.9 \pm 0.4$ |
| FlowGT | $\mathbf{98.5 \pm 0.2}$ | $96.0 \pm 1.3$ | $98.9 \pm 0.6$ | $98.3 \pm 0.7$ |

Table 2: Multiclass classification results for the cascading failure analysis on the PowerGraph dataset using three MPLs for all models. Reported results represent the balanced accuracy on the test set in %, averaged over five training runs with different random seeds, along with the corresponding standard deviation. The best result for each test system is marked in bold.

| Model | IEEE24 | IEEE39 | IEEE118 | UK |
|---|---|---|---|---|
| GCN | $90.8 \pm 0.7$ | $82.7 \pm 3.3$ | $83.6 \pm 4.8$ | $89.0 \pm 1.5$ |
| GIN | $97.1 \pm 0.9$ | $94.7 \pm 2.2$ | $98.4 \pm 1.4$ | $\mathbf{98.4 \pm 0.5}$ |
| GAT | $92.1 \pm 3.4$ | $84.7 \pm 7.9$ | $92.7 \pm 1.2$ | $94.6 \pm 1.2$ |
| GATv2 | $93.7 \pm 1.6$ | $88.2 \pm 3.8$ | $93.4 \pm 1.4$ | $88.4 \pm 13.6$ |
| GT | $96.7 \pm 0.9$ | $92.3 \pm 1.9$ | $98.7 \pm 0.4$ | $96.1 \pm 0.7$ |
| FlowGAT | $92.0 \pm 2.6$ | $93.1 \pm 1.0$ | $97.0 \pm 1.0$ | $93.7 \pm 3.0$ |
| FlowGATv2 | $96.7 \pm 0.9$ | $\mathbf{95.7 \pm 0.7}$ | $\mathbf{98.9 \pm 0.5}$ | $96.8 \pm 0.9$ |
| FlowGT | $\mathbf{98.4 \pm 0.5}$ | $94.1 \pm 0.8$ | $98.3 \pm 0.7$ | $97.3 \pm 0.5$ |

### 4.3 PREDICTING PROPERTIES OF OPERATIONAL AMPLIFIERS

**Experimental setting** We train three versions of each model on the prediction of the Op-Amp properties gain, bandwidth, and FoM, respectively. Thereby, we split the dataset into train/validation/test with ratios 80/10/10% and select the same test set as in Dong et al. (2023) for better comparison. We use the AdamW optimizer (Loshchilov, 2017) with an initial learning rate of $10^{-4}$ and train each model using the mean squared error (MSE) as the loss function with a batch size of 64 for a maximum of 500 epochs but apply early stopping with a patience of 20 epochs. For the general GNNs, we use two message-passing layers with a hidden dimension of 301 combined with a ReLU activation as well as a global mean pooling operator for readout. For the DAG models (D-VAE, DAGNN, PACE), we use the default parameters from Dong et al. (2023) and the model-specific readout layers. For DAFlowGNN, we train a single-layer and a two-layer variant (DAFlowGNN-1, DAFlowGNN-2) and adopt all other model parameters from DAGNN. Since one DAFlowGNN-layer contains twice as many model parameters compared to a DAGNN-layer, we train a two-layer and a four-layer-DAGNN (DAGNN-2, DAGNN-4) for a fair comparison. The final prediction is done using a two-layer perceptron with a ReLU activation in between. Right before these final layers, we apply a dropout of 50% for regularization purposes. Each individual training run is repeated ten times with different random seeds.

**Discussion** The RMSEs on the test set for all models and all OpAmp target properties are presented in Tab. 3. From the standard message-passing GNNs, GIN and GT perform the best, showing significantly lower prediction errors compared to GCN and GAT. GATv2 performs equally well on the prediction of bandwidth and FoM but shows an increased RMSE on gain. However, all of these models yield higher prediction errors compared to the directed acyclic GNNs (PACE, D-VAE, and

Table 3: Regression results for the prediction of three different Op-Amp properties from the Ckt-Bench101 dataset. Reported results represent the RMSE on the test set in %, averaged over ten training runs with different random seeds, along with the corresponding standard deviation. The best result for each property is marked in bold.

| Model | Gain | Bandwidth | FoM |
|---|---|---|---|
| GCN | $0.485 \pm 0.081$ | $0.570 \pm 0.012$ | $0.578 \pm 0.028$ |
| GIN | $0.281 \pm 0.007$ | $0.455 \pm 0.008$ | $0.450 \pm 0.007$ |
| GAT | $0.425 \pm 0.027$ | $0.590 \pm 0.049$ | $0.565 \pm 0.046$ |
| GATv2 | $0.324 \pm 0.011$ | $0.458 \pm 0.020$ | $0.440 \pm 0.009$ |
| GT | $0.271 \pm 0.008$ | $0.440 \pm 0.024$ | $0.439 \pm 0.018$ |
| FlowGAT | $0.334 \pm 0.088$ | $0.470 \pm 0.054$ | $0.462 \pm 0.049$ |
| FlowGATv2 | $0.340 \pm 0.043$ | $0.474 \pm 0.020$ | $0.485 \pm 0.020$ |
| FlowGT | $0.405 \pm 0.050$ | $0.432 \pm 0.016$ | $0.429 \pm 0.010$ |
| PACE | $0.253 \pm 0.009$ | $0.443 \pm 0.014$ | $0.443 \pm 0.009$ |
| D-VAE | $0.218 \pm 0.003$ | $0.426 \pm 0.005$ | $0.425 \pm 0.007$ |
| DAGNN-2 | $0.216 \pm 0.002$ | $0.396 \pm 0.006$ | $0.396 \pm 0.009$ |
| DAGNN-4 | $0.210 \pm 0.003$ | $0.394 \pm 0.008$ | $0.394 \pm 0.006$ |
| DAFlowGNN-1 | $0.215 \pm 0.003$ | $0.388 \pm 0.004$ | $0.387 \pm 0.005$ |
| DAFlowGNN-2 | $\mathbf{0.209 \pm 0.007}$ | $\mathbf{0.371 \pm 0.008}$ | $\mathbf{0.366 \pm 0.008}$ |

DAGNN), which leverage the sequential nature of DAGs, resulting in significant performance boosts across all target properties.

Applying FlowGAT, FlowGATv2, and FlowGT to the OpAmps yields mixed results when comparing them to their standard counterparts GAT, GATv2, and GT. While FlowGAT performs significantly better than GAT, FlowGATv2, and FlowGT do not show any significant improvements but rather perform worse compared to GATv2 and GT, especially in predicting the gain. The likely reason for this is that although the flow attention weights account for resource conservation in flow graphs, the computed flow attention weights might not be meaningful, since they are conditioned on ancestor nodes only. Here, classical attention weights might lead to more expressive models in some cases. Furthermore, these FlowGNNs do not process DAGs sequentially according to their partial order and instead are restricted to aggregate information from only a $k$-hop node neighborhood, where $k$ is the number of MPLs. This explains the significantly higher RMSEs compared to directed acyclic GNNs. DAFlowGNN solves both of these problems by leveraging the sequential nature of DAGs *and* computing meaningful flow attention weights $\beta_{ij}$, which are conditioned on both ancestors of node $j$ and descendants of node $i$. Consequently, the two-layer variant of this model shows the best performance on all target properties among all tested models, including DAGNN-4, which exhibits the same degree of complexity as DAFlowGNN-2. Similarly, DAFlowGNN-1 also shows lower prediction errors compared to DAGNN-2 and even performs better than DAGNN-4 on two of the three target properties.

## 5 CONCLUSION

In this paper, we proposed FlowGNN, a GNN framework based on a flow attention mechanism that accounts for the conservation of resources in flow graphs. We also extended this framework to DAGs, resulting in a model called DAFlowGNN, which can distinguish non-isomorphic directed acyclic flow graphs which were so far indistinguishable for existing GNNs tailored to DAGs. Furthermore, we showed that our models outperform their standard counterparts on graph-level regression and classification tasks across different flow graph datasets. In the future, it could be interesting to analyze how the proposed models scale to larger circuits and power grids. Another interesting direction for future work would be to investigate the performance of the proposed models on node- and edge-level tasks, as well as on other flow graph data, such as traffic networks or supply chains. Additionally, more theoretical work is required to gain a deeper understanding of the expressivity of our models compared to standard GNNs.

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

## A  APPENDIX

### A.1  SCORING FUNCTIONS OF ATTENTIONAL GNN BASELINES

In GAT (Veličković et al., 2018), the scoring function is defined as

$$e_{\text{GAT}}\left(\boldsymbol{h}_i, \boldsymbol{h}_j\right) = \text{LeakyReLU}\left(\boldsymbol{a}^T \cdot \left[\boldsymbol{W}\boldsymbol{h}_i \,\|\, \boldsymbol{W}\boldsymbol{h}_j\right]\right). \tag{16}$$

Thereby, the linear layers $\boldsymbol{a}$ and $\boldsymbol{W}$ are applied consecutively, making it possible to collapse them into a single linear layer.

In GATv2 (Brody et al., 2022), a strictly more expressive attention mechanism is proposed, in which the second linear layer $\boldsymbol{a}$ is applied *after* the nonlinearity:

$$e_{\text{GATv2}}\left(\boldsymbol{h}_i, \boldsymbol{h}_j\right) = \boldsymbol{a}^T \text{LeakyReLU}\left(\boldsymbol{W} \cdot \left[\boldsymbol{h}_i \,\|\, \boldsymbol{h}_j\right]\right). \tag{17}$$

Thus, GATv2 is effectively using a multi-layer perceptron (MLP) to compute the attention scores, allowing for *dynamic* attention compared to the *static* attention performed by GAT.

Finally, GT (Shi et al., 2021) is transferring the attention mechanism of the Transformer model (Vaswani et al., 2017) to graph learning:

$$\boldsymbol{q}_i = \boldsymbol{W}_q \boldsymbol{h}_i + \boldsymbol{b}_q, \tag{18}$$

$$\boldsymbol{k}_j = \boldsymbol{W}_k \boldsymbol{h}_j + \boldsymbol{b}_k, \tag{19}$$

$$e_{\text{GT}}\left(\boldsymbol{h}_i, \boldsymbol{h}_j\right) = \frac{\boldsymbol{q}_i^T \cdot \boldsymbol{k}_j}{\sqrt{d}}, \tag{20}$$

where $\boldsymbol{q}_i \in \mathbb{R}^d$ is the query vector, $\boldsymbol{k}_j \in \mathbb{R}^d$ is the key vector and $\boldsymbol{W}_q, \boldsymbol{W}_k, \boldsymbol{b}_q, \boldsymbol{b}_k$ are trainable parameters.

All of the above scoring functions can be extended to multi-head attention and are able to incorporate edge features as well. This characteristic is naturally inherited by the corresponding FlowGNNs.

### A.2  DIRECTED ACYCLIC GNN BASELINES

In the encoder of the D-VAE model (Zhang et al., 2019), the aggregation corresponds to a gated sum using a mapping network $m$ and a gating network $g$, and the update function $\phi$ is a gated recurrent unit (GRU) (Cho et al., 2014):

$$\boldsymbol{m}_i' = \sum_{j \in \mathcal{N}_{in}(i)} g(\boldsymbol{h}_j') \odot m(\boldsymbol{h}_j'), \tag{21}$$

$$\boldsymbol{h}_i' = \text{GRU}(\boldsymbol{h}_i, \boldsymbol{m}_i'). \tag{22}$$

Table 4: Number of nodes and edges for each test system as well as the number of corresponding graph samples contained in the PowerGraph dataset (see Varbella et al. (2024)).

| Test system | No. Nodes | No. Edges | No. Graphs |
|---|---|---|---|
| IEEE24 | 24 | 38 | 21500 |
| IEEE39 | 39 | 46 | 28000 |
| IEEE118 | 118 | 186 | 122500 |
| UK | 29 | 99 | 64000 |

Table 5: Distribution of the classification labels for each test system in the PowerGraph dataset (see Varbella et al. (2024)). DNS stands for *demand not served* and c. f. stands for *cascading failure*, corresponding to at least one more tripping branch after the initial outage.

| Test system | **Category A**
DNS > 0 MW
c. f. | **Category B**
DNS > 0 MW
no c. f. | **Category C**
DNS = 0 MW
c. f. | **Category D**
DNS = 0 MW
no c. f. |
|---|---|---|---|---|
| IEEE24 | 15.8% | 4.3% | 0.1% | 79.7% |
| IEEE39 | 0.55% | 8.4% | 0.45% | 90.6% |
| IEEE118 | >0.1% | 5.0% | 0.9% | 93.9% |
| UK | 3.5% | 0% | 3.8% | 92.7% |

Another popular model is the DAGNN (Thost & Chen, 2021), which also uses a GRU for the update function but the message function is an attention mechanism with model parameters $\boldsymbol{w}_1$ and $\boldsymbol{w}_2$:

$$\boldsymbol{m}'_i = \sum_{j \in \mathcal{N}_{in}(i)} \alpha_{ij} \left( \boldsymbol{h}_i, \boldsymbol{h}'_j \right) \boldsymbol{h}'_j, \tag{23}$$

$$\alpha_{ij} = \operatorname*{softmax}_{j \in \mathcal{N}_{in}(i)} \left( \boldsymbol{w}_1^{\mathrm{T}} \boldsymbol{h}_i + \boldsymbol{w}_2^{\mathrm{T}} \boldsymbol{h}'_j \right). \tag{24}$$

Since the embeddings of the (possibly multiple) end nodes contain information on the whole DAG, they are typically used for computing the graph-level representations. After $L$ layers, the graph-level embedding can be obtained by concatenating the end node representations from all layers followed by a max-pooling across all end nodes:

$$\boldsymbol{h}_{\mathcal{G}} = \operatorname*{Max\text{-}Pool}_{i \in \mathcal{F}} \left( \Big\|_{l=0}^{L} \boldsymbol{h}_i^l \right). \tag{25}$$

### A.3 DETAILS ON POWERGRAPH AND CKT-BENCH101

The PowerGraph dataset contains four different test systems (IEEE24, IEEE39, IEEE118, UK) with unique graph structures. For the cascading failure analysis, each test system was simulated for different power grid loading conditions together with a specific initial outage, resulting in a large number of graph samples. The number of nodes and edges in each test system as well as the number of graph samples are reported in Tab. 4.

Tab. 5 shows how the classification labels are distributed in the PowerGraph dataset for each test system. For multiclass classification, models are trained to distinguish all available categories, while for binary classification, the models only have to predict whether DNS > 0 MW (categories A and B) or DNS = 0 MW (categories C and D), where DNS is the demand not served. Due to the strong class imbalance, the balanced accuracy BA is used as the evaluation metric (Brodersen et al., 2010), which is defined as the mean of sensitivity and specificity:

$$\mathrm{BA} = \frac{1}{2} \left( \frac{\mathrm{TP}}{\mathrm{TP} + \mathrm{FN}} + \frac{\mathrm{TN}}{\mathrm{TN} + \mathrm{FP}} \right). \tag{26}$$

Here, TP/FP/TN/FN represent true/false positive/negative predictions.

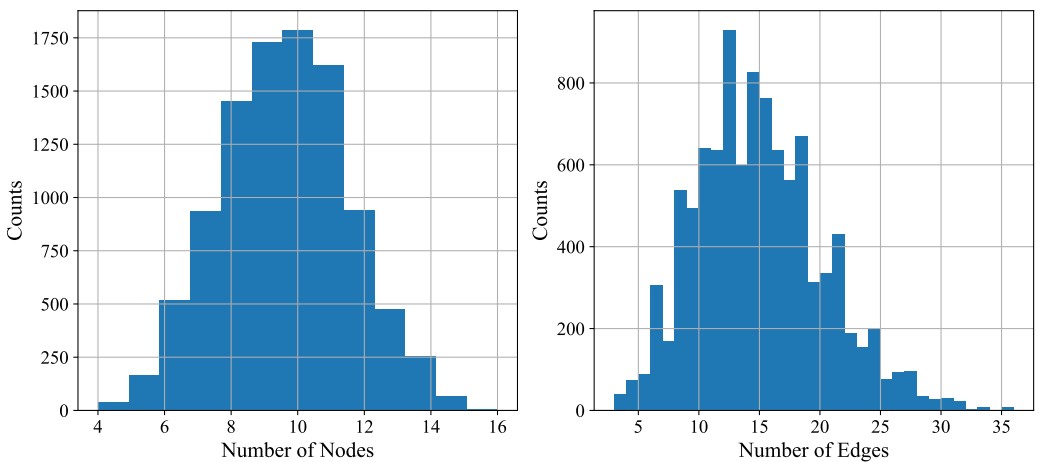

Figure 4: Distribution of the number of nodes (left) and number of edges (right) within the Ckt-Bench101 dataset (Dong et al., 2023).

Table 6: Binary classification results for the cascading failure analysis on the PowerGraph dataset using a single MPL for all models. Reported results represent the balanced accuracy on the test set in %, averaged over five training runs with different random seeds, along with the corresponding standard deviation. The best result for each test system is marked in bold.

| Model (1 layer) | IEEE24 | IEEE39 | IEEE118 | UK |
|---|---|---|---|---|
| GCN | $70.7 \pm 3.3$ | $67.8 \pm 1.5$ | $71.9 \pm 1.6$ | $81.0 \pm 2.3$ |
| GIN | $92.0 \pm 2.1$ | $89.6 \pm 1.1$ | $94.0 \pm 10.4$ | $\mathbf{97.8 \pm 0.6}$ |
| GAT | $82.7 \pm 2.2$ | $66.9 \pm 2.3$ | $71.9 \pm 1.6$ | $86.6 \pm 0.3$ |
| GATv2 | $86.5 \pm 2.5$ | $73.2 \pm 1.6$ | $75.1 \pm 1.6$ | $90.3 \pm 6.2$ |
| GT | $85.4 \pm 2.8$ | $66.8 \pm 3.8$ | $71.9 \pm 1.6$ | $90.8 \pm 2.0$ |
| FlowGAT | $92.4 \pm 1.8$ | $78.5 \pm 2.7$ | $75.5 \pm 1.5$ | $97.2 \pm 0.6$ |
| FlowGATv2 | $94.3 \pm 1.3$ | $86.8 \pm 0.8$ | $\mathbf{99.2 \pm 0.3}$ | $96.3 \pm 0.7$ |
| FlowGT | $\mathbf{95.4 \pm 0.7}$ | $\mathbf{92.4 \pm 0.9}$ | $99.1 \pm 0.2$ | $97.3 \pm 0.3$ |

The CktBench-101 dataset from the Open Circuit Benchmark Dong et al. (2023) contains 10,000 artificially generated operational amplifiers represented as DAGs. Fig. 4 shows the distribution of the number of nodes and the number of edges among all graphs in the dataset. The average number of nodes is $9.6$ with a standard deviation of $2.1$. The average number of edges is $14.5$ with a standard deviation of $5.3$. We are using the most recent update of the CktBench-101 dataset, which does not contain any failed simulations anymore.

## A.4 ADDITIONAL RESULTS FOR THE CASCADING FAILURE ANALYSIS

In addition to the results for the cascading failure analysis reported in Tab. 1 and Tab. 2, where each model contains three MPLs, we also performed similar experiments for one and two MPLs, respectively. These results are reported in Tab. 6- 9. Interestingly, the increase in performance of FlowGNNs compared to their standard counterparts is more pronounced for single-layer GNNs. However, the overall performance drops when using fewer MPLs for almost all models, test systems, and tasks.

## A.5 EFFICIENCY COMPARISON

We compare the average training and inference times for processing the training set of the Ckt-Bench101 dataset, which contains 8,000 OpAmps graphs. Thereby, we use the same parameters

Table 7: Binary classification results for the cascading failure analysis on the PowerGraph dataset, using two MPLs for all models. Reported results represent the balanced accuracy on the test set in %, averaged over five training runs with different random seeds, along with the corresponding standard deviation. The best result for each test system is marked in bold.

| Model (2 layers) | IEEE24 | IEEE39 | IEEE118 | UK |
|---|---|---|---|---|
| GCN | $87.4 \pm 1.8$ | $78.2 \pm 1.0$ | $73.7 \pm 1.9$ | $85.7 \pm 4.0$ |
| GIN | $\mathbf{97.5 \pm 1.0}$ | $\mathbf{96.9 \pm 1.2}$ | $99.4 \pm 0.4$ | $\mathbf{98.3 \pm 0.7}$ |
| GAT | $94.1 \pm 2.7$ | $75.4 \pm 8.0$ | $81.7 \pm 2.1$ | $97.9 \pm 0.2$ |
| GATv2 | $91.4 \pm 2.9$ | $89.4 \pm 4.1$ | $89.6 \pm 10.6$ | $97.5 \pm 0.7$ |
| GT | $94.1 \pm 1.0$ | $82.4 \pm 6.1$ | $79.3 \pm 1.1$ | $98.0 \pm 0.1$ |
| FlowGAT | $96.3 \pm 0.2$ | $95.5 \pm 1.1$ | $99.4 \pm 0.4$ | $\mathbf{98.3 \pm 0.4}$ |
| FlowGATv2 | $96.2 \pm 0.8$ | $93.3 \pm 2.9$ | $\mathbf{99.5 \pm 0.2}$ | $97.7 \pm 0.3$ |
| FlowGT | $96.9 \pm 2.1$ | $95.1 \pm 1.0$ | $99.0 \pm 0.5$ | $98.0 \pm 0.3$ |

Table 8: Multiclass classification results for the cascading failure analysis on the PowerGraph dataset, using a single MPL for all models. Reported results represent the balanced accuracy on the test set in %, averaged over five training runs with different random seeds, along with the corresponding standard deviation. The best result for each test system is marked in bold.

| Model (1 layer) | IEEE24 | IEEE39 | IEEE118 | UK |
|---|---|---|---|---|
| GCN | $58.0 \pm 1.3$ | $65.9 \pm 1.8$ | $67.3 \pm 1.5$ | $61.8 \pm 0.7$ |
| GIN | $92.0 \pm 4.9$ | $86.5 \pm 3.5$ | $89.7 \pm 10.2$ | $\mathbf{96.3 \pm 1.8}$ |
| GAT | $76.0 \pm 1.8$ | $63.7 \pm 3.1$ | $68.1 \pm 1.9$ | $76.0 \pm 1.0$ |
| GATv2 | $84.0 \pm 3.4$ | $68.5 \pm 3.3$ | $73.5 \pm 1.9$ | $83.5 \pm 7.1$ |
| GT | $79.0 \pm 3.1$ | $68.6 \pm 2.9$ | $68.2 \pm 1.6$ | $83.4 \pm 4.3$ |
| FlowGAT | $89.7 \pm 3.1$ | $74.4 \pm 3.4$ | $76.0 \pm 1.2$ | $95.5 \pm 1.2$ |
| FlowGATv2 | $92.5 \pm 1.5$ | $83.6 \pm 1.3$ | $97.0 \pm 0.9$ | $89.2 \pm 2.1$ |
| FlowGT | $\mathbf{93.5 \pm 0.5}$ | $\mathbf{88.5 \pm 0.8}$ | $\mathbf{98.1 \pm 0.4}$ | $93.3 \pm 0.9$ |

that we reported in Sec. 4.3. The results for general GNNs and FlowGNNs are reported in Fig. 5. We observe that the non-attentional GNNs (GCN and GIN) are slightly more efficient compared to the attentional GNNs. Furthermore, we can not observe any significant increases in training or inference time for FlowGNNs compared to their standard counterparts. The reason for this is that the only modification of the FlowGNNs is the different normalization of the attention scores in the flow attention mechanism, which does not affect the model's efficiency.

Fig. 6 shows a similar efficiency comparison for the directed acyclic GNNs. These models are significantly more expensive to compute compared to general GNNs due to the sequential message-passing (DVAE, DAGNN, and DAFlowGNN) or a much higher number of model parameters (PACE). We observe slight differences in efficiency between these models. However, they are much harder to compare because we used original implementations from the authors rather than standard implementations from PyTorch Geometric (Fey & Lenssen, 2019), as in the case of the general GNNs. Note that the efficiency of PACE can be considerably increased through parallelization (Dong et al., 2022), which is not reflected in this analysis.

All experiments were carried out on NVIDIA V100 GPUs.

Table 9: Multiclass classification results for the cascading failure analysis on the PowerGraph dataset, using two MPLs for all models. Reported results represent the balanced accuracy on the test set in %, averaged over five training runs with different random seeds, along with the corresponding standard deviation. The best result for each test system is marked in bold.

| Model (2 layers) | IEEE24 | IEEE39 | IEEE118 | UK |
|---|---|---|---|---|
| GCN | $79.4 \pm 8.9$ | $74.6 \pm 1.5$ | $70.6 \pm 1.3$ | $70.2 \pm 6.1$ |
| GIN | $\mathbf{97.1 \pm 0.6}$ | $\mathbf{95.2 \pm 1.5}$ | $97.3 \pm 2.3$ | $\mathbf{97.9 \pm 0.4}$ |
| GAT | $87.0 \pm 6.4$ | $75.7 \pm 3.7$ | $78.5 \pm 3.0$ | $93.7 \pm 0.8$ |
| GATv2 | $90.5 \pm 1.0$ | $88.2 \pm 1.0$ | $96.2 \pm 1.3$ | $92.7 \pm 2.0$ |
| GT | $91.0 \pm 1.5$ | $82.4 \pm 5.1$ | $74.4 \pm 2.4$ | $94.5 \pm 0.6$ |
| FlowGAT | $94.7 \pm 1.3$ | $94.8 \pm 1.2$ | $\mathbf{98.9 \pm 0.4}$ | $96.8 \pm 0.4$ |
| FlowGATv2 | $96.0 \pm 1.1$ | $92.0 \pm 1.6$ | $97.2 \pm 0.9$ | $95.6 \pm 0.4$ |
| FlowGT | $96.4 \pm 0.7$ | $92.2 \pm 1.5$ | $98.2 \pm 0.7$ | $96.9 \pm 0.6$ |

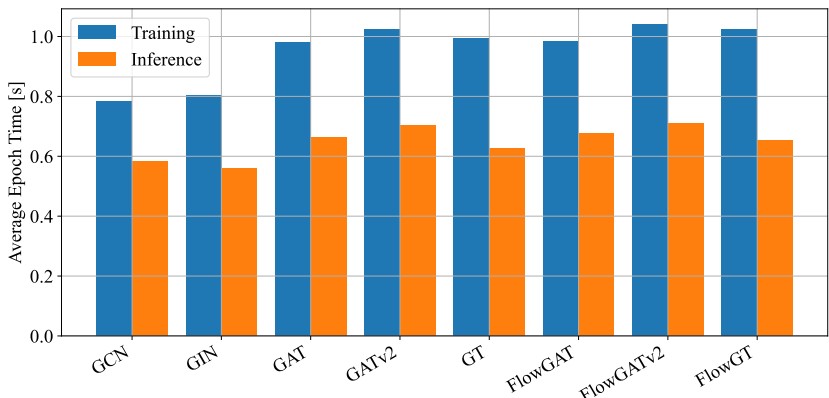

Figure 5: Training and inference times of different GNN and FlowGNN models for processing the whole training set (8,000 graphs) from the CktBench101 dataset. Thereby, a batch size of 64 is used.

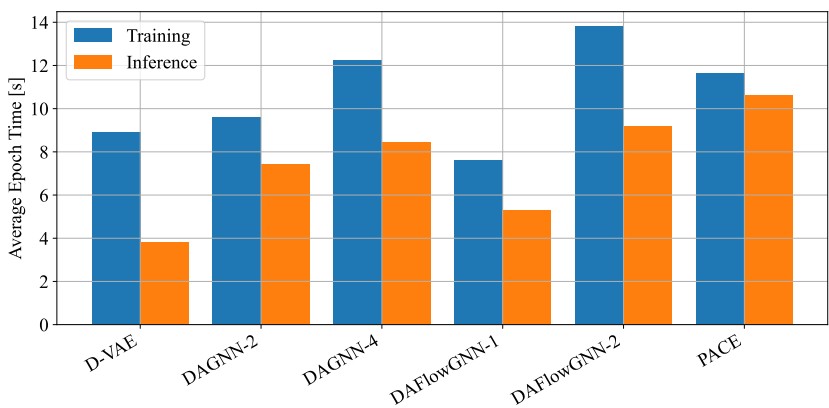

Figure 6: Training and inference times of different directed acyclic GNN models for processing the whole training set (8,000 graphs) from the CktBench101 dataset. Thereby, a batch size of 64 is used.

