# OpenReview forum: "Flow Graph Neural Networks"
_ICLR.cc/2025/Conference — ICLR 2025 Conference Withdrawn Submission_

### Official Review · Reviewer_rWJv · 2024-10-18

**Soundness:** 2
**Presentation:** 3
**Contribution:** 2
**Rating:** 3
**Confidence:** 3

**Summary:**

This paper proposes a novel framework for graph neural networks (GNNs) called FlowGNN to handle conservation laws in graph-structured data involving resource flows, such as electrical grids and traffic networks. Different from existing GNN methods, FlowGNN employs a new flow attention mechanism that aggregates messages across outgoing neighbors to respect resource conservation. Furthermore, this framework is extended to Directed Acyclic Graphs (DAGs) and can distinguish non-isomorphic flow graphs. Experiments on the power grid and electronic circuit datasets demonstrate the superior performance of FlowGNN in graph-level classification and regression tasks.

**Strengths:**

- FlowGNN modifies the traditional attention mechanism by normalizing across outgoing neighbors instead of incoming neighbors. This simple yet impactful change addresses the limitations of traditional attention mechanisms in modeling physical flows.
- Additionally, FlowGNN extends existing GNN frameworks to DAGs, which enables it to distinguish non-isomorphic flow graphs that standard GNNs cannot differentiate.

**Weaknesses:**

- The writing of this paper could be improved. For instance, the introduction is overly long (about 2 pages) and lacks clarity, which makes it difficult for readers to understand the key motivations of the work.

- The theoritical analysis of FlowGNN is limited. This paper can not provide a comprehensive exploration of the expressiveness and computational complexity of FlowGNN compared to standard GNNs. Although the practical effectiveness of FlowGNN are demonstrated through experiments, a deeper theoretical understanding of how the proposed flow attention mechanism help FlowGNN handle resource conservation and distinguish non-isomorphic flow graphs is needed.

- The novelty of FlowGNN is limited. This paper primarily focuses on modifying the attention mechanism by normalizing across outgoing neighbors instead of incoming ones. However, this adjustment does not introduce new architectures or theoretical innovations beyond those found in existing methods. Additionally, the extension of FlowGNN to DAGs represents only an incremental advancement.

- The experimental results are not convincing, and the baseline methods are not strong enough. Although the authors compare FlowGNN method with some widely used baseline methods like GCN, GAT, GIN, GT and PACE, these are neither the latest nor the most competitive, which undermines the significance of the reported performance.

**Questions:**

- The authors should clearly explain why FlowGNN fails to achieve state-of-the-art (SOTA) performance in the binary classification task while achieving SOTA in the multiclass classification task compared to baseline methods.
- The time and space complexity of the FlowGNN method should also be provided in the methodology section, which is necessary for comparison with the traditional graph attention mechanism.
- More recent DAG methods should be introduced in the experimental part for fair comparisons, such as GraphNOTEARS [1], ContextualDAG [2], and DAG [3].

[1] Fan S, Zhang S, Wang X, et al. Directed acyclic graph structure learning from dynamic graphs[C] Proceedings of the AAAI Conference on Artificial Intelligence. 2023, 37(6): 7512-7521.

[2] Thompson R, Bonilla E V, Kohn R. Contextual directed acyclic graphs[C]//International Conference on Artificial Intelligence and Statistics. PMLR, 2024: 2872-2880.

[3] Luo Y, Thost V, Shi L. Transformers over directed acyclic graphs[J]. Advances in Neural Information Processing Systems, 2024, 36.

---

### Official Review · Reviewer_DWbc · 2024-10-22

**Soundness:** 2
**Presentation:** 2
**Contribution:** 3
**Rating:** 5
**Confidence:** 4

**Summary:**

The authors tackle the expressiveness problem of recursive neural networks and GNNs when these are applied to a specific family of real-world graphs. In fact, it might be important to model the conservation of “information flow” in graphs like power grids and electronic circuits, and to this end the authors introduce an architectural bias that is reminiscent of graph attention. The new mechanism normalizes each incoming message according to the outgoing edges of the source node, rather than based on the incoming set of edges for the target node. This way, the contribution of each node is not duplicated during message-passing.
In addition, they propose a recursive network tailored to DAGs (DAFlowGNN) that incorporates said bias. The authors show that DAFlowGNN can distinguish non-isomorphic DAGs where recent models specifically tailored to DAGs fail.

**Strengths:**

The paper is generally well written and presented, figures are clear, neat, and well supported by captions. The organization feels a bit off, mostly due to the alternating between the new attention mechanism and the DAG architecture throughout the paper methods and experiments, but the intentions and message of the authors remain clear.

It deserves mention the fact that the paper is self-contained and strives to get one simple message across, namely that we could “invert” the way we compute edge weights in attention mechanisms to prevent duplication of the same information across multiple outgoing edges. The technique introduced by authors is properly motivated and its scope is well defined.

The paper is also very detailed in the experimental section, and it seems that one might be able to easily replicate the results provided in the tables. The empirical comparison considers a reasonable number of baselines and datasets, considering all models have been retrained and a small model selection has been run. Despite results on the PowerGraph dataset do not look particularly exciting, I believe the Ckt-Bench101 is the dataset where the architectural bias of the proposed architecture might provide convincing evidence (although there are some concerns to be addressed).

A simple discussion hints at DAFlowGNN being able to discriminate graphs that other recent neural networks for DAGs cannot, although it is unclear the influence of the normalization operation compared to the reverse pass (see questions). I believe the contribution lies at the intersection between recursive architectures for Directed Positional Acyclic Graphs (DPAGs) and the recent architectures for DAGs (that need to generalize without making any causal assumption on the parent nodes).

Overall, this paper proposes a simple but interesting architectural change that has much sense in specific contexts and I believe it can help to advance our understanding of the field.

**Weaknesses:**

It appears that the paper mentions several times the concept of “conservation of flow”, e.g., Korchhoff’s law, but my understanding is that there is no guarantee or mention that the flow is conserved at all: nodes will receive a certain amount of message information (greater or smaller of a “full single message”) and will always output a “single” message to its outgoing edges, that is without duplicating information. There might be something I am missing, but if this is really the case then the impact of the paper would be greatly reduced. As a matter of fact, if flow is not conserved then it is not surprising at all that the GIN method has similar performances to FlowGAT on the PowerGraph benchmark, as both the sum aggregator and the flow attention weights are susceptible to degree changes (while GCN for instance would be susceptible to distributional changes in the neighborhood).

Similarly, the claims that the authors need to distinguish “all” non-isomoprhic DAGs (lines 265-266) might be misleading. For instance, let us focus on flow attention weights and not on DAFlowGNN. What happens if you take the circuit in Fig 1.b left and you compare it with “in” -> branch into R1 x2 -> merge into R2 --> out? If you do not take into account normalization for incoming edges, you might end up with the same result also with FlowGAT. In any case, despite the claim made in said lines, the authors only (partially) showed that DAFlowGNN could distinguish a particular pair of DAGs, not that it is universal with respect to all DAGs. To summarize, some claims should be clarified/adapted to reflect the real contributions of the paper and avoid general statements.

Despite its novelty, the paper fails to acknowledge and position itself compared to a whole line of less recent works on recursive and contextual architectures that are relevant because they often incorporate the same architectural bias than recent works. Before GNNs were applied to DAGs -- in reverse order compared to historical developments -- (contextual) recursive neural networks for DAGs [1,2] and their expressive power were completely characterized [3], and later the causality assumptions were relaxed to allow cycles in the computation [4, Scarselli et al. 2009].

Finally, there are some concerns on empirical results. The authors have correctly split the dataset into training/validation/test. The validation set should be used to select the best hyper-parameters for a specific model on a specific dataset (a process called “model selection”). Then, once the best hyper-parameter configuration has been found on the validation set, the model should be (possibly after retraining) evaluated on the unseen test set. Given the specific hold-out strategy adopted in the paper, it makes no sense to show results for all models with a certain number of layers, e.g., DAGNN-2/4 or DAFlowGNN-1/2. What the authors should show is the estimate of the risk using the best configuration attained by each model and dataset. Arguments like the ones made at lines 415-417 cannot stand against the correct process for risk estimation. In this regard, the authors are recommended to show results attained by each model using only the best configuration found on the validation set. This would allow a fair comparison of different *classes of* models rather than a comparison of simple point estimates that will likely be misleading. The authors can refer to slides 23-33 of Samy Bengio’s lecture: https://bengio.abracadoudou.com/lectures/theory.pdf.

[1] Sperduti, Alessandro, and Antonina Starita. "Supervised neural networks for the classification of structures." IEEE transactions on neural networks 8.3 (1997): 714-735.

[2] Frasconi, Paolo, Marco Gori, and Alessandro Sperduti. "A general framework for adaptive processing of data structures." IEEE transactions on Neural Networks 9.5 (1998): 768-786.

[3] Hammer, Barbara, Alessio Micheli, and Alessandro Sperduti. "Universal approximation capability of cascade correlation for structures." Neural Computation 17.5 (2005): 1109-1159.

[4] Micheli, Alessio. "Neural network for graphs: A contextual constructive approach." IEEE Transactions on Neural Networks 20.3 (2009): 498-511.

**Questions:**

-	Could the authors comment on the absence of conservation of flow and how this impacts the discussions throughout the paper? Do you perhaps agree that the paper and the naming of the techniques might be changed to prevent the reader from being misled that the “flow” is indeed conserved?
-	The authors could show more convincingly that DAFlowGNN can discriminate the two DAGs of Figure 3. One (better) way would be to prove it slightly more formally and precisely in the paper, showing what are the benefits of running the reverse pass compared to the forward pass only. An alternative would be to run an ablation study to show it on synthetic graphs. Could  you comment on this?
-	Could the authors provide a revised table of numbers where only the best performing configuration on the validation set is used to provide a risk estimate on the test set? Provided that model selection was already performed, it would amount to re-evaluate just the best configuration for each model and dataset. This would allow us to fairly compare and evaluate the benefit of DAFlowGNN on the Ckt-Bench101.
-	Not really a comment, but it would help to refer to specific figures 1.a-1.b in the text rather than Fig 1.

---

### Official Review · Reviewer_JQHA · 2024-10-31

**Soundness:** 3
**Presentation:** 2
**Contribution:** 2
**Rating:** 5
**Confidence:** 3

**Summary:**

The paper introduces FlowGNNs, a framework designed to improve GNNs for flow graphs where resource conservation is essential. FlowGNNs adapt attention mechanisms to prevent arbitrary duplication of node information by normalizing outgoing messages, thus enforcing resource conservation. Additionally, the paper presents DAFlowGNN, which extends FlowGNNs to directed acyclic graphs (DAGs), enabling the differentiation of non-isomorphic flow graphs that standard DAG-specific GNNs cannot distinguish. Experiments on electronic circuits and power grid data show FlowGNNs’ and DAFlowGNN’s superiority over traditional GNNs in classification and regression tasks.

**Strengths:**

1. This paper proposes a new flow graph neural network to address the constraint of resource conservations.
2. The framework is extended to DAGs, which enable discrimination between non-isomorphic flow graphs.

**Weaknesses:**

1. The novelty of this paper seems limited, as author just normalize the attention scores across outgoing neighbors instead of across incoming neighbors. The novelty of this paper needs to illustrate better. Specifically, for non-expert readers, it is unclear how significant impact such change leads to.

2. The relationship between FlowGNN and DAFlowGNN is unclear. While DAFlowGNN is a DAG version of FlowGNN, it is unclear what the differences of two GNNs in terms of applicability, e.g.  Are they both applicable to flow graphs?  For their architectures, it is unclear what are the main differences.

3.  In Section 3.3, it is better to illustrate in terms of rigor theorems  that DAFlowGNN is more expressive than any DAGNN, as the current version is not easy to follow.

4. For the experiment section, electronic circuits and power grids both belong to  electrical domain, which makes it doubtful that the proposed GNNs may have narrow applications limited by resource conservations. Can authors show more applications and compare results? Another concern is the computational expenses. As author admit in the paper, DAFlowGNN is twice as computationally expensive as DAGNN, but the efficiency experiments are missing, and it is necessary to discuss what techniques can be used to reduce computational time.

**Questions:**

Please refer to questions mentioned in the weakness sections.

---

### Official Review · Reviewer_Tzqj · 2024-11-04

**Soundness:** 3
**Presentation:** 3
**Contribution:** 3
**Rating:** 6
**Confidence:** 4

**Summary:**

The authors propose Flow Graph Neural Networks (FlowGNNs) and their extension to directed acyclic graphs with the target of ensuring the conservation laws associated with the flow of physical resources, in certain graphs (flow graphs). Their idea is very simple: normalize outgoing messages instead of incoming ones during message passing in GNNs. Doing so (flow attention), duplication, which would be a conservation violation, is avoided. To extend this simple idea to directed acyclic graphs (DAFlowGNN), this normalization step as part of a forward pass, follows a propagation stage of hidden states from nodes back to their predecessors (reverse pass). In this way all nodes of the graph connected with some given node are taken into account in computing its representation. FlowGNNs are shown to be more expressive than other GNNs for DAGs. In the experiments, they equip attention-based architectures (GAT, GT) with the proposed outgoing-message normalization (i.e. FlowGAT, FlowGT) and apply them for binary and multiclass graph classification tasks on power grid datasets and for graph regression tasks (i.e. predict values of properties) for a graph dataset of operational amplifiers. FlowGNN-based architectures are generally more performant (respectively: balanced accuracy and RMSE metrics) over popular GNNs (or DAG-based GNNs), closely followed by GIN-based and DAGNN models.

**Strengths:**

Very simple and intuitive idea: also its motivation by emphasizing the difference of informational and flow graphs immediately captures reader's interest.

**Weaknesses:**

- Performance results in some cases for FlowGNNs are mixed and the benefits of the approach are not directly realized in the experiments (Table 3). In other cases, a simple GNN like GIN can be more performant (Table 1). So there is room for improvement in the consistency of the gains reported.

- Typically the notation (u, v) for a directed edge is assumed to denote a connection from node u to node v, not the other way around as in here.

**Questions:**

Intuition behind reverse pass in DAFlowGNN should be revisited: it is not completely clear why it is needed, particularly why "flow of some arbitrary physical resource from node j to node i in a flow graph may also depend on all descendants of the node i," (lines 272-273)?

---

### Official Review · Reviewer_eu21 · 2024-11-04

**Soundness:** 2
**Presentation:** 2
**Contribution:** 2
**Rating:** 3
**Confidence:** 4

**Summary:**

In the submitted manuscript the authors pose the problem of Graph Neural Networks (GNNs), designed to study flows on graphs, being unable to distinguish non-isomorphic graphs. The authors then go on to propose a variation of the attention mechanism of the Graph Attention Network (GAT) in which attention scores are not normalised over incoming, but rather outgoing edges, which means that the resulting GNN adheres to Kirchoff's first law of information preservation. They furthermore extend the DAGNN, to the DAFlowGNN, which now includes both a forward and a backward pass through the directed acyclic graph. Subsequently, they offer some theoretical discussion of what graphs may be distinguishable by their DAFlowGNNs. Finally, their FlowGNNs and DAFlowGNNs are shown to outperform several baseline models on the majority of the studied datasets.

**Strengths:**

- The idea of enforcing Kirchhoff's first law in GNNs is interesting.
- The methodological propositions in this paper, i.e., the normalisation over outgoing edges and both a forward and backward pass on DAGs, are sensible.
- In practice, the proposed methods appear to work somewhat well.

**Weaknesses:**

Please find further details on my listed weaknesses in my questions below.
- The empirical evidence in favour of your model is not very strong, in the sense that you are sometimes outperformed by the Graph Isomorphism Network (published in 2019) and some improvements are marginal.
- The proposed model is not described in sufficient detail, i.e., not all aspects of your proposed model are well-motivated, the notation is not always clearly defined and certain models such as the FlowGT are not discussed at all.
- The theoretical component of this work is not formal and rather small. This would be fine if there was overwhelming empirical evidence in favour of the proposed method, which does not seem to be the case either. So, I want to suggest to the authors to formalise and to further extend their theoretical study.

**Questions:**

1] The following statement in Line 83 "such as GCN (Kipf & Welling, 2017) [...] In these models, messages exchanged between neighboring nodes do not depend on the number of message recipients." appears to be false. In the GCN the messages of neighbours are normalised by the square rooted node degrees of both nodes involved in the edge. Therefore, the number of message recipients is taken into account in the weighting of messages in the GCN. In fact, it seems to me that the normalisation scheme of the GCN considering both outgoing and incoming number of neighbours could give rise to an interesting extension of your FlowGAT model, although it would violate Kirchhoff's law.

2] Your proposed FlowGNN is not described in sufficient detail:

2.1] Since you now represent each node via two hidden representations $h^{rv}$ and $h^{fw}$ is unclear to me what the hidden states $h_i$ in Equations (9), (10) and (11) refer to.

2.2] It is unclear how the hidden states $h^{rv},$ $h^{fw}$ and $h_i$ are initialised.

2.3] The attention scheme you use in your FlowGNNs is not the standard GAT attention scheme since you do not have the LeakyReLU and one less weight matrix in the attention scheme (see Equations (16) and (17)). This difference should be better motivated, since you discuss the GAT model as a corresponding model to your FlowGAT model.

2.4] It is unclear to me why you use a GRU model on the sequence $h^{rv}, m^{fw}$ of length two and do not use an MLP in which these are concatenated.

2.5] Similarly it is unclear how the other variants of your model, e.g., the FlowGT, are computed. It would be great if you could provide the associated model equation. I am particularly curious about the FlowGT model since it is unclear to me how forward and backward message passing works in a model that freely rewires the entire graph in its attention scheme.

3] In Section 3.3 you provide a promising start to a theoretical result on the expressivity of your model. It would however be more concise if you could formalise your discussion in a Theorem such that your theoretical claims are more easily to identify and more precisely proofed.

4] It seems to me that also your experimental analysis could be improved:

4.1] It is customary in many papers to experiment with models of different depths, i.e., with varying numbers of layers, and to report the results for the optimal number of layers as established in a grid search. I recommend that you take this approach to your experiments and in particular, experiment with deeper models, since your maximal number of message passing layers appears to be optimal and it remains unclear whether even deeper models would perform better.

4.2] I am unsure why you use the global maximum pooling (see Line 409). On graph level tasks often sum pooling is observed to be optimal. Could you please motivate this seemingly unusual choice?


5] Minor comment: In Lines 272-4 your sentence "However, since the flow of some arbitrary physical resource from node $j$ to node $i$ in a flow graph may also depend on all descendants of the node $i$" begs the question to me, why you do not consider bidirectional edges in these cases. Answering this question in your manuscript would strengthen the paper in my view.

---

### Note · Authors · 2024-11-19

**Comment:**

We have decided to withdraw our paper. We would like to thank all reviewers for their time and constructive feedback.

**Withdrawal Confirmation:**

I have read and agree with the venue's withdrawal policy on behalf of myself and my co-authors.